# Engineering self-deliverable ribonucleoproteins for genome editing in the brain

Kai Chen [1,2,9], Elizabeth C. Stahl[1,2,3,9], Min Hyung Kang[1,2,4], Bryant Xu[1,2], Ryan Allen[1,2], Marena Trinidad[1,2,4] & Jennifer A. Doudna [1,2,3,4,5,6,7,8] ✉

The delivery of CRISPR ribonucleoproteins (RNPs) for genome editing in vitro and in vivo has important advantages over other delivery methods, including reduced off-target and immunogenic effects. However, effective delivery of RNPs remains challenging in certain cell types due to low efficiency and cell toxicity. To address these issues, we engineer self-deliverable RNPs that can promote efficient cellular uptake and carry out robust genome editing without the need for helper materials or biomolecules. Screening of cell-penetrating peptides (CPPs) fused to CRISPR-Cas9 protein identifies potent constructs capable of efficient genome editing of neural progenitor cells. Further engineering of these fusion proteins establishes a C-terminal Cas9 fusion with three copies of A22p, a peptide derived from human semaphorin-3a, that exhibits substantially improved editing efficacy compared to other constructs. We find that self-deliverable Cas9 RNPs generate robust genome edits in clinically relevant genes when injected directly into the mouse striatum. Overall, self-deliverable Cas9 proteins provide a facile and effective platform for genome editing in vitro and in vivo.

CRISPR-Cas technology, adapted from bacterial immune systems, has been a revolutionary molecular tool for genome engineering in cells and organisms[1,2]. Guided by an RNA component (sgRNA), the Cas protein uses its endonucleolytic activity to create a double-stranded break at a target genomic site, inducing site-specific DNA repair that introduces sequence changes into the genome. This RNA-guided activity is also used to introduce site-specific sequence changes (base edits) and reverse transcriptase-generated sequence changes (prime edits)[2]. CRISPR-Cas9 genome editing technology is already being used to treat genetic disorders[3,4]. However, the expanded use of CRISPR-Cas technology as a therapy will require efficient and safe delivery methods to transport the editing molecules into target cells in vivo[5]. In

particular, delivering CRISPR-Cas complexes to the central nervous system (CNS) remains a big challenge[6]. Current brain delivery methods include intracranial injection of viral vectors[7], such as adeno-associated virus (AAV) encoding Cas proteins and corresponding sgRNAs. However, viral delivery can be immunogenic and induce undesired insertional mutagenesis, and the production of viral vectors faces manufacturing difficulties[8,9]. In contrast, the direct delivery of CRISPR-Cas ribonucleoproteins (RNPs), if available, could avoid drawbacks of viral delivery systems[10–15].

Delivery of CRISPR-Cas RNPs to the brain is of particular interest due to its potential to induce genome edits that could be protective against neurodegeneration. Non-viral delivery strategies have

[1]Department of Molecular and Cell Biology, University of California, Berkeley, Berkeley, CA, USA. [2]Innovative Genomics Institute, University of California, Berkeley, CA, USA. [3]California Institute for Quantitative Biosciences, University of California, Berkeley, Berkeley, CA, USA. [4]Howard Hughes Medical Institute, University of California, Berkeley, Berkeley, CA, USA. [5]Gladstone Institutes, San Francisco, CA, USA. [6]Gladstone-UCSF Institute of Genomic Immunology, San Francisco, CA, USA. [7]Molecular Biophysics and Integrated Bioimaging Division, Lawrence Berkeley National Laboratory, Berkeley, CA, USA. [8]Department of Chemistry, University of California, Berkeley, Berkeley, CA, USA. [9]These authors contributed equally: Kai Chen, Elizabeth C. Stahl. ✉e-mail: doudna@berkeley.edu

employed different types of nanoparticles to encapsulate CRISPR-Cas9 or -Cas12a RNPs, as exemplified by the "CRISPR-Gold" system[16], nano-complexes with amphiphilic peptides[17], PEGylated nanocapsules[18], and glucose-conjugated silica nanoparticles[19]. These examples have shown that different cell types can be edited with variable efficiencies, some of which were sufficient to improve disease pathology in mouse models[16,17]. However, these delivery methods rely heavily on nano-particle optimization and manufacturing, which may restrict their therapeutic applications. Alternatively, cell-permeable Cas9 RNPs could provide a simpler and more broadly applicable strategy for genome editing both in vitro and in vivo. Peptides with the ability to penetrate the cell membrane could be used to endow Cas9 with self-deliverable capability. Previous work demonstrated that Cas9 RNPs fused with multiple positively charged nuclear localization sequences (SV40 NLSs) are capable of self-delivery to mouse neurons for genome editing purposes[20,21]. Recent works have shown that synthetically designed endosomolytic peptides can effectively promote Cas9 RNP delivery to primary cells in vitro in a non-covalent manner[22,23]. We thus hypothesized that engineering genetic fusions of Cas9 and highly functional CPPs would help us establish robust RNPs with (targeted) self-delivery capability, especially for in vivo genome editing.

Here we describe the development of cell-permeable CRISPR RNPs for delivery to neural progenitor cells in vitro and to neurons

in vivo. Screening and systematic engineering of Cas9 fused to different cell-penetrating peptides identified a potent construct capable of effective self-delivery and genome editing. The use of this self-delivery approach to edit clinically relevant genes in the mouse brain demonstrates the potential utility of this method for treating genetic disorders in the CNS.

## Results

### Comparison of SpyCas9 and LbCas12a for in vitro and in vivo genome editing

We first evaluated two widely used genome editing enzymes, Cas9 from *Streptococcus pyogenes* (SpyCas9) and Cas12a from *Lachnospiraceae bacterium ND2006* (LbCas12a), for self-delivery and genome editing efficiency when fused to nuclear localization signal (NLS) peptides (Fig. 1a). The Cas genome editors together with corresponding sgRNA or crRNA were designed to edit a stop cassette sequence and initiate expression of a tdTomato fluorescent protein in mouse-derived neural progenitor cells (NPCs) (Fig. 1b). The Cas12a used in this study (iCas12a[24]) has improved genome editing activity relative to the wild-type LbCas12a. We tested two versions of SpyCas9 and iCas12a with different copy numbers of an NLS tethered to the N or C terminus of the proteins to induce delivery into NPCs. The ribonucleoproteins (RNPs) were assembled and added to cell cultures at

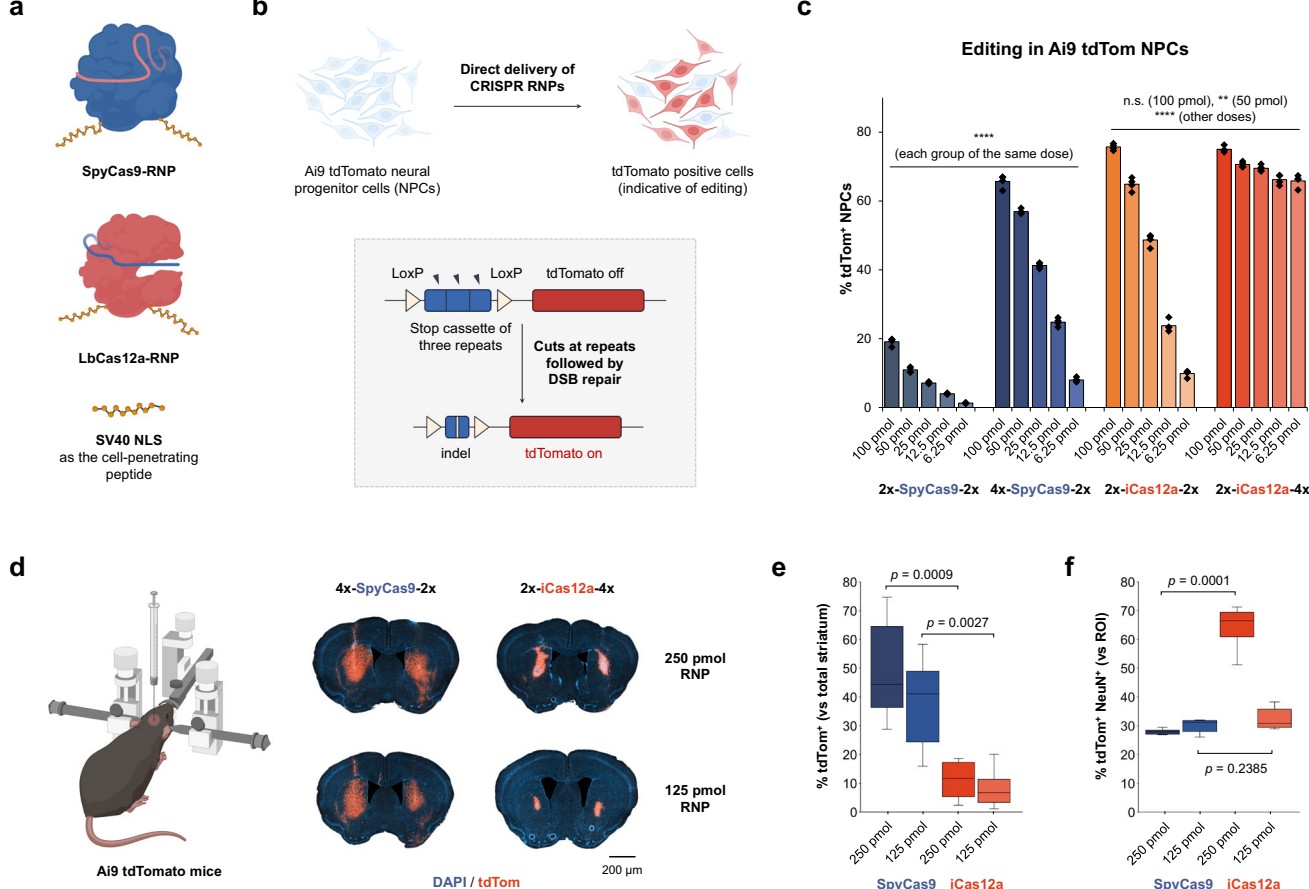

**Fig. 1 | Comparison between SpyCas9 and LbCas12a for cell-penetrating SV40 NLS-assisted delivery and gene editing in neural cells in vitro and in vivo.**
**a** Graphic illustrations of two RNP constructs. **b** Schematic of Cas9 or Cas12a RNP-mediated editing of Ai9 tdTomato NPCs to turn on fluorescent signals.
**c** Quantification of tdTom⁺ NPCs based on the direct delivery of gene editors. $n = 4$ for each group, data are presented as mean values with individual data points.
**d** Comparison of the gene editing activities of cell-permeable RNPs based on SpyCas9 and iCas12a in Ai9 mouse brain. **e** Editing volumes in the striatal tissue based on the injection of different RNP dosages. $n = 6$ for each group. **f** Co-

expression of tdTomato and NeuN quantified per regions of interest (ROIs), e.g., edited area per hemisphere. $n = 4$ for the group of iCas12a (250 pmole), $n = 5$ for the other groups. Statistical analyses (unpaired $t$ test) were performed for (**c, e, f**) and $p$ values (**$p < 0.01$, ****$p < 0.0001$, and n.s. - not significant) were indicated with each set of quantification. Data are presented in box plots for (**e, f**) where the lower bound of the lower whisker shows the minimum, the lower bound of the box shows the lower quartile, the center of the box shows the median, the upper bound of the box shows the upper quartile and the upper bound of the upper whisker shows the maximum. Images in (**a, b, d**) were created with biorender.com.

different dosages, and genome editing levels were quantified five days post-delivery using flow cytometry based on the tdTomato signal. Consistent with previous observations[21], Cas9 or Cas12a with more NLS copies at the protein termini (6 vs. 4 copies in total) had improved delivery capability. Interestingly, the Cas12a protein with 2 and 4 NLS copies at the N and C termini, respectively, showed highly efficient delivery and robust genome editing in vitro even with a relatively low RNP dosage (av. 66% editing observed with 6.25 pmol RNP, cal. 50 nM RNP in cell culture) (Fig. 1c).

To assess genome editing in the brain using these cell-permeable RNPs, we tested their in vivo editing activities in mice. We performed intraparenchymal injections of the RNPs at two different dosages (250 pmol and 125 pmol) into the striatum of Ai9 mice (Fig. 1d). Fourteen days after RNP injection, the Ai9 mice were sacrificed, and brains were isolated for analysis. We observed tdTomato⁺ cells in the striatum with both genome editors and at both RNP dosages. However, SpyCas9 outperformed iCas12a by 3 to 6 folds when we compared the total edited striatal volume (Fig. 1e). On the other hand, iCas12a RNP edited more neurons (NeuN⁺) within the tdTomato⁺ region of interest (ROI) per hemisphere than SpyCas9 RNP (Fig. 1f). These results suggest that SpyCas9 RNPs diffuse more readily from the injection site in the brain, whereas the edits performed by iCas12a RNPs are more localized around the injection site.

## Screening CPPs for improved cell-penetrating ability of SpyCas9 RNPs

With SpyCas9 being a better enzyme than iCas12a for genome editing in the brain, we wanted to further engineer the Cas9 protein for self-delivery by fusing it to different cell-penetrating peptides (CPPs). Categorized into different classes according to their native functions, origins, and other properties, CPPs have been intensively investigated for their ability to transport small-molecule drugs, nucleic acids, and proteins into cells[25,26]. To evaluate the capability of these peptides to deliver Cas9 RNPs to neural progenitor cells, we fused different CPPs to the C-terminus of a Cas9 protein bearing two copies of an NLS at the N terminus (Fig. 2a). We selected 34 CPPs with different chemical features (Fig. 2b; Supplementary Table 1). To facilitate the screening of different fusion proteins, we redesigned the bacterial expression constructs with two tags, CL7[27] and His₆ tags at the N- and C-terminus, respectively, which allowed for rapid purification of fusion proteins in parallel (Fig. 2c). Protein purification using two gravity columns based on nickel-NTA and Im7-6B resins yielded most of the desired proteins in high purity (>90%) and good quantity (3–10 mg purified proteins per liter of bacterial culture) without the need for ion-exchange or size-exclusion chromatography.

The purified fusion constructs were then evaluated for their self-delivery capability in Ai9 tdTomato NPCs (Fig. 2d). The RNP bearing two NLS copies at the N- and C-termini of Cas9 (i.e., 2x-Cas9-2x, x stands for one copy of SV40 NLS) was used as the standard reference in the screening. Screening identified nine peptides with improved delivery capability compared to two NLS copies alone. Interestingly, these functional CPPs can all be classified as cationic peptides, while anionic or hydrophobic CPPs tested in this study were ineffective for Cas9 RNP delivery. Nevertheless, some cationic peptides (e.g., Penetratin, EB1, LALF, etc.) failed to give good genome editing outcomes in NPCs based on CPP-mediated RNP delivery. Nucleofection control experiments suggested that these peptide fusions to the Cas9 protein disrupted its capacity to induce genome editing in cells. Among all the CPPs tested, antimicrobial peptide Bac7[28], heparin-binding peptide HBP[29], HIV-derived peptide CA-Tat[30], and a C-terminal peptide derived from semaphorin-3a, A22p[31], represent the most effective peptides for RNP delivery to NPCs, producing >3-fold improvements in genome editing compared to the reference 2x-Cas9-2x RNP in the tdTomato NPC assay.

## Optimization of SpyCas9-CPP fusions and mechanistic profiles of CPP-assisted delivery

Having identified promising CPPs for RNP delivery, we wanted to further engineer these Cas9-CPP fusion constructs for higher delivery efficiency. The most effective peptide, Bac7, and the least cationic peptide among the top four candidates, A22p, were chosen. Engineering of the corresponding fusion constructs involved systematic changes to the peptide fusion location, altering the combination of CPP and NLS sequences, and testing different copy numbers of CPPs (Fig. 3a). The screening of different CPPs suggested that human c-myc NLS (in three copies, with six net positive charges) could function similarly to the SV40 NLS (in two copies, with ten net positive charges), and thus human c-myc NLS was also examined in RNP constructs.

We first explored the effect of peptide location on RNP delivery efficacy (Fig. 3b) using the tdTomato NPC assay. When peptide Bac7 was fused to the N-terminus of Cas9 bearing an SV40 NLS at the C-terminus (construct #2), a subtle decrease in delivery efficiency was observed. Interestingly, with human c-myc NLS fused at the C-terminus, the N-terminal Bac7 construct (#4) showed a mildly enhanced genome editing level of 58% compared to other Bac7-based constructs. However, the N-terminal A22p constructs (#8 and #10) were less effective for RNP delivery driven by the CPP. In our protein purification strategy, a C-terminal peptidic sequence S3 (SLEVLFQ) used for HRV 3 C protease recognition to cleave purification tags is always left over, and we wondered whether this sequence could inhibit the exposure of the CPP and then RNP delivery (Fig. 3c). Constructs with CPPs fully exposed at the C-terminus were prepared and tested for delivery. No major difference was observed for the Bac7-fused constructs (#1 and #5) in delivery; however, the A22p-fused construct boosted genome editing from 40% (construct #7) to 55% (construct #11) when the C-terminal peptide is fully exposed. These results together suggest that the location of the Bac7 peptide at the N- or C-terminus has minimal impact on its activity, whereas the A22p peptide prefers a fully exposed C-terminal fusion. A22p is a C-terminal peptide in human semaphorin-3a, consistent with its preferred location at the Cas9 C-terminus. We then tested whether the Bac7 peptide can be inserted into the solvent-exposed region in the Cas9 backbone to drive RNP delivery (Fig. 3d)[32]. After testing different insertion sites and optimizing the linkers for Bac7 insertion (Supplementary Fig. 4), construct #6 bearing two copies of the SV40 NLS at the N- and C-termini of the protein and two copies of Bac7 at site 205 in the protein backbone was found to preserve Cas9's genome editing activity when delivered using nucleofection (Supplementary Fig. 5). However, the self-delivery efficiency of the insertion construct is much worse than that observed for the terminal fusion constructs.

To explore how CPPs promote RNP delivery, we tested RNP delivery with CPPs provided in trans (Fig. 3e). The reference RNP construct based on 2x-Cas9-2x was mixed with 10 equivalents of synthetic CPPs (in single copy) prior to treatment of Ai9 NPCs. Interestingly, we observed 1.5 to 2.5-fold improvements in genome editing with the reference RNP mixed with Bac7 or A22p compared to that without additional peptides; some other CPPs are less effective in promoting RNP delivery in trans (Supplementary Fig. 6). Recent studies have shown that amphipathic peptides can be used to promote the delivery of Cas9 or Cas12a RNP in trans by facilitating endocytosis[22,23]. We thus rationalized that the Bac7 or A22p peptides may perform similar functions. To elucidate the delivery mechanism, we employed different endocytosis inhibitors in RNP delivery experiments. Treatment of NPCs with endocytosis inhibitors, monensin (80 nM) or cytochalasin D (1.2 μM), two hours before RNP delivery led to a substantial decrease in genome editing by 4 to 5 folds (Supplementary Fig. 8a)[33]. We also observed that treating cells with growth factors could enhance editing levels based on direct delivery of Cas9 RNPs (Supplementary Fig. 8b, c).

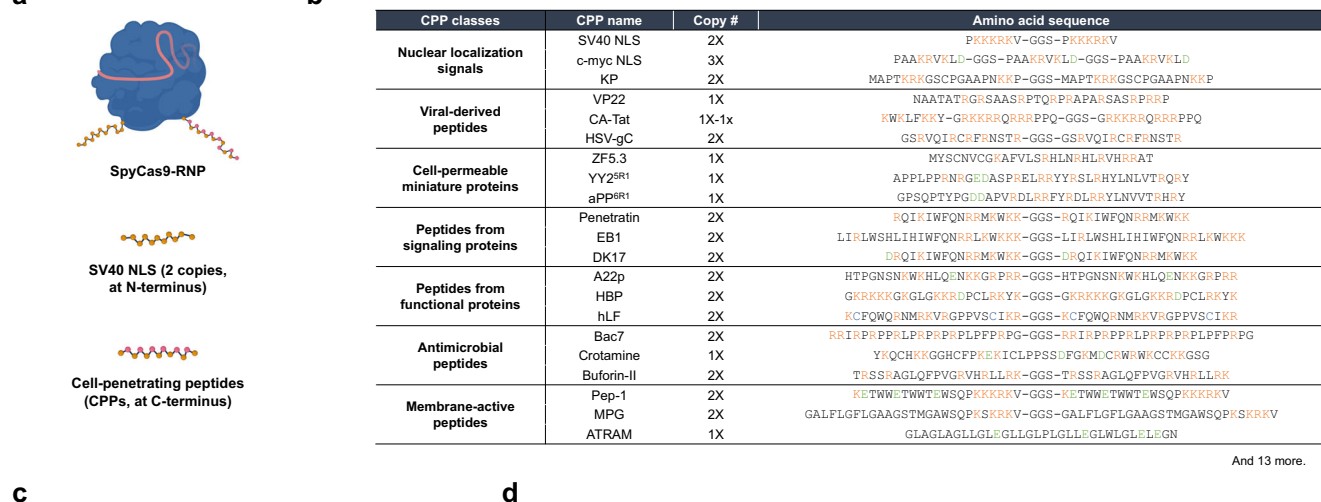

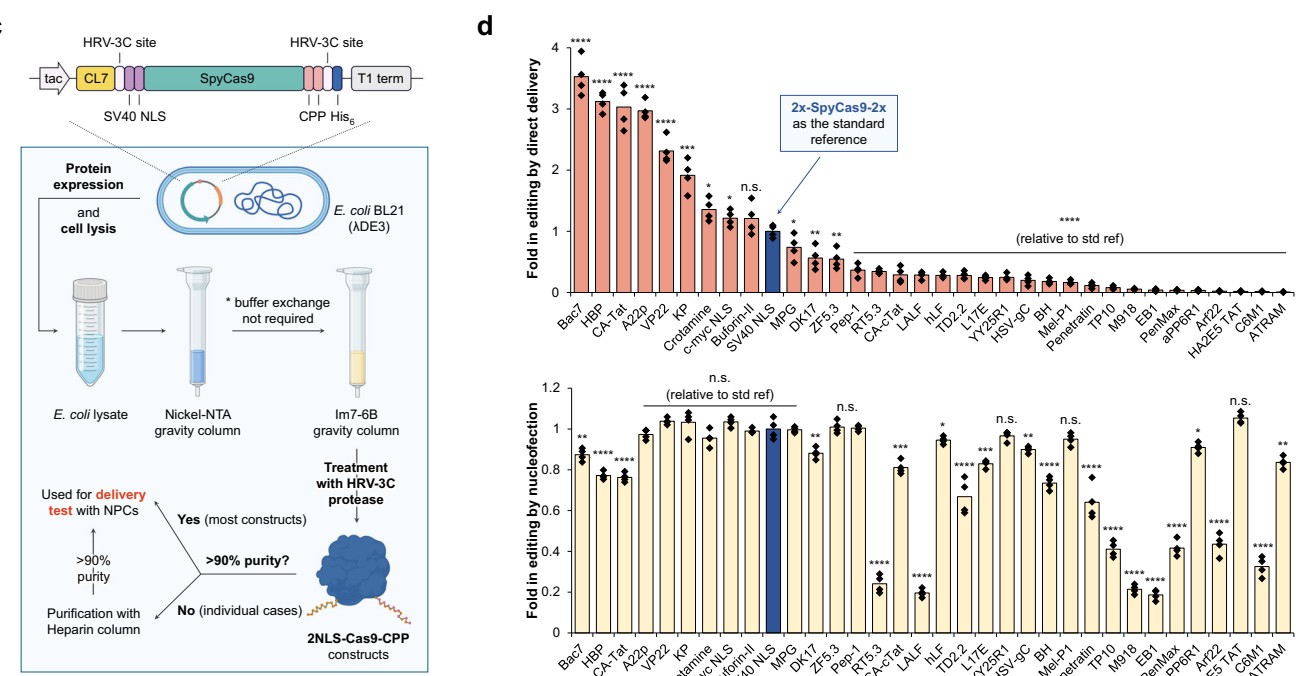

**Fig. 2 | Screening of different cell-penetrating peptides to evaluate their ability in cellular delivery of SpyCas9 RNPs. a** Graphic illustrations of Cas9 RNP constructs fused with CPPs. **b** Sequence information of representative CPPs used in the screening. **c** Schematic of the expression and purification systems for Cas9-CPP fusion proteins in the screening. **d** Screening of Cas9-CPP constructs for self-delivery and gene editing activities using Ai9 tdTom NPCs. Fold change in tdTom⁺ NPCs% based on (upper) direct delivery and (lower) nucleofection of RNPs with 2x-SpyCas9-2x as the standard reference. 100 pmol RNP used for each NPC editing experiment. $n = 4$ for each group, data are presented as mean values with individual data points. Statistical analyses (unpaired $t$ test) were performed, and $p$ values (*$p < 0.1$, **$p < 0.01$, ***$p < 0.001$, ****$p < 0.0001$, and n.s. - not significant) were indicated with each set of quantification. Images in (**a**, **c**) were created with biorender.com.

We next tested whether more copies of CPPs on Cas9 can improve RNP self-delivery efficiency (Fig. 3f). We found that three copies of the A22p peptide (A22p3) at the C-terminus (construct #12) boosted the genome editing efficiency from 40% (by two copies of A22p, construct #7) to 70% in turning on tdTomato with NPCs. However, the protein stability dropped substantially, as observed by protein aggregation during purification, when more copies of A22p (copy number ≥4) were fused to Cas9, and the Bac7 peptide in three copies on Cas9 was highly toxic to *E. coli* due to its antimicrobial nature[34] and resulted in poor expression of the corresponding construct. Removing the proteolytically left-over peptide from the C terminus of the A22p3 construct (#13) did not result in a further increase in RNP delivery, giving a comparable editing efficiency of 70% to construct #12. In addition, as cysteines on the protein surface could form disulfide bonds between

molecules leading to protein aggregation, we mutated the two surface cysteine residues on Cas9 to serine and generated a mutant Cas9, Cas9(dS), with its genome editing function fully preserved. The final construct #16 with three copies of the A22p peptide fully exposed at the C-terminus of Cas9(dS) gave an average of 72% genome editing efficiency based on direct RNP delivery to NPCs to turn on tdTomato.

**Evaluation of SpyCas9-A22p3* constructs for genome editing in the mouse brain**

With the Cas9-A22p construct established for effective delivery to neuronal cells, we wanted to assess its ability for in vivo delivery and genome editing. We first compared the engineered construct (2x-Cas9(dS)-A22p3*) with the previously tested NLS-enriched construct (4x-Cas9-2x) for their self-delivery capability in vitro and in vivo. In

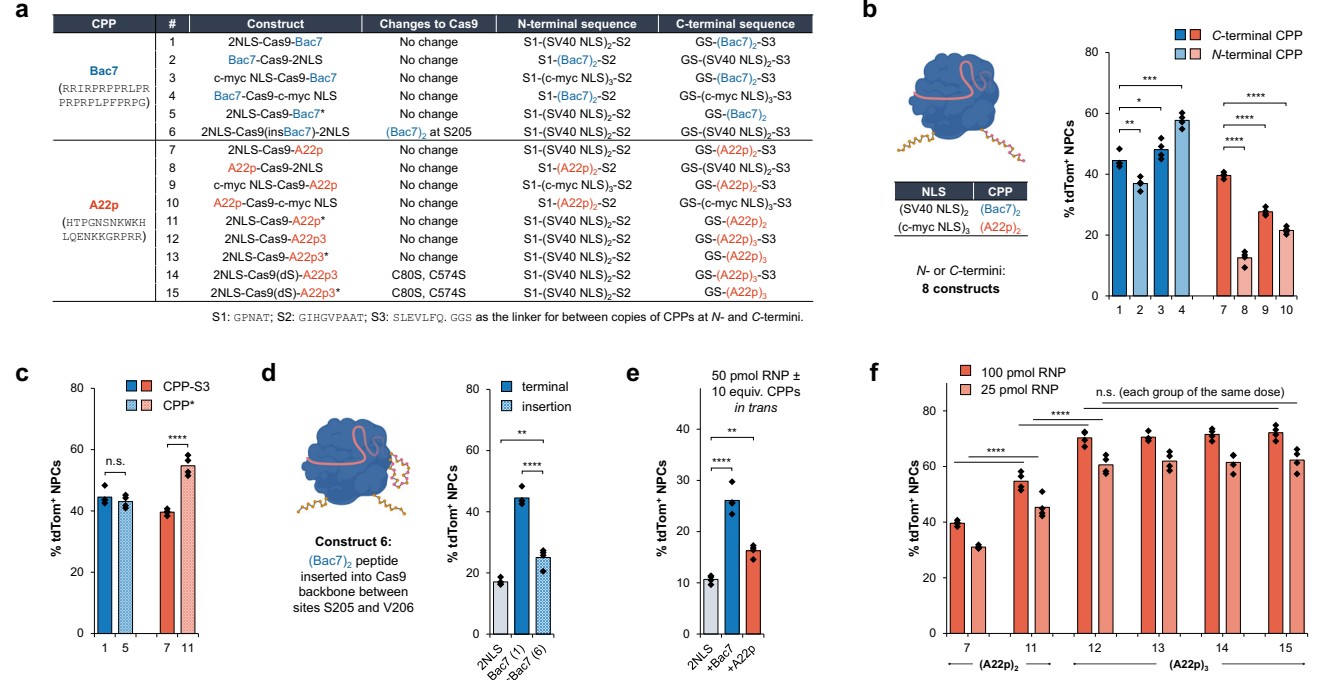

**Fig. 3 | Further optimization of SpyCas9-CPP fusions for improved delivery efficacy and mechanistic profiles of CPP-assisted delivery.** The bar graphs in this figure present the quantification of tdTom+ NPCs based on the direct delivery of corresponding RNPs. 100 pmol RNP used for each NPC editing experiment, unless specified. **a** Table of Cas9-Bac7/A22p constructs (* indicates removal of S3). **b** Locational effects of the peptides at Cas9 termini in combination with different NLS peptides. **c** Terminal exposure effect of the peptides. **d** Insertion of Bac7 peptide in Cas9 backbone. **e** CPP-promoted RNP delivery in trans. **f** Systematic engineering of Cas9-A22p constructs. $n = 4$ for each group, data are presented as mean values with individual data points for (**b**–**f**). Statistical analyses (unpaired $t$ test) were performed for (**b**–**f**), and $p$ values (*$p < 0.1$, **$p < 0.01$, ***$p < 0.001$, ****$p < 0.0001$, and n.s. - not significant) were indicated with each set of quantification. Images in (**b**, **d**) were created with biorender.com.

vitro tests suggested that the engineered construct could be effectively delivered to NPCs and produced robust genome editing even with relatively low doses of RNPs (av. 47% editing with 12.5 pmol RNP, cal. 100 nM RNP in cell culture), showing nearly two-fold higher efficiency than the 4x-Cas9-2x construct (Fig. 4a). Following in vitro tests, we next evaluated the genome editing efficacy of the engineered construct in the brain using Ai9 mice (Fig. 4b). Robust genome editing was observed with the whole series of A22p3*-RNP doses from 250 pmol to 25 pmol (i.e., 50 μM to 5 μM, 5 μl/injection), while the 4x-Ca9-2x RNP was less effective with low doses of RNP (e.g., 75 pmol and 25 pmol; i.e., 15 μM and 5 μM, 5 μl/injection), especially when considering the percentage of edited neurons within the tdTomato+ ROI (Fig. 4c, d).

The high self-delivery efficacy of the engineered Cas9 RNP in vivo motivated further experiments to determine genome editing efficiencies at endogenous genomic sites relevant to central nervous system (CNS) disorders. As a proof of concept, we selected two genes encoding tyrosine hydroxylase (TH)[35] and metabotropic glutamate receptor 5 (mGluR5)[36], respectively, which are associated with the pathogenesis of neurological disorders including Parkinson's disease (PD) and Alzheimer's disease (AD) (Fig. 5a). We first tested whether these two genes can be disrupted in neural progenitor cells. Direct treatment of NPCs with A22p3*-conjugated genome editors (12.5 or 25 pmol, cal. 100 or 200 nM RNP in cell culture) targeting the two genes produced robust genome editing with efficiencies up to 72%, as quantified by next-generation sequencing (NGS) (Fig. 5b). To test in vivo genome editing efficacy, Cas9 RNPs (250 pmol; i.e., 50 μM, 5 μL/injection) were injected into the mouse striatum. Two weeks after RNP injection, mouse striatums were dissected for NGS and qPCR analyses to quantify the genome editing efficiency at the DNA and mRNA levels. The NGS results revealed an average of 1.5–5% editing levels for TH and

mGluR5 (Fig. 5c), resulting in an average of 15–20% reduction in the expression level of TH and mGluR5, as quantified by qPCR (Fig. 5d). These results suggest the potential utility of self-deliverable RNPs for genome editing in the brain, at least in a small animal model system.

## Discussion

Delivery of CRISPR genome editors in the form of RNPs provides transient editing activity that minimizes off-target effects and immune reactions caused by viral or nanoparticle delivery systems[14,37]. In addition, direct RNP delivery can lead to effective genome editing in cells with low transcription and translation activities, as exemplified by embryonic or tissue stem cells[38]. Compared to delivery strategies involving nanoparticles, the development of cell-permeable RNPs capable of self-delivery provides a facile and potentially cost-effective strategy for genome editing in vitro and in vivo. In particular, genome editing in the brain using direct injection of cell-permeable RNPs has minimal immunogenic effects when the editing materials are produced with high purity[21]. In this study, RNPs fused with CPPs display higher delivery and editing efficiencies in the brain compared to NLS-coupled Cas9 RNPs[20].

Cell-penetrating peptides with the ability to cross cell membranes nondestructively have been used for the delivery of various therapeutic reagents. The strategy of using CPPs to drive CRISPR RNP delivery has been tested for genome editing in cultured cells[39–41]. An early example demonstrated that chemical conjugation of Cas9 protein with a poly-arginine-based CPP enabled genome editing in human cell lines with reduced off-target effects compared to plasmid DNA transfection[36]. However, CPP-based RNP delivery has been less widely tested for in vivo genome editing[42], presumably due to the limited RNP self-delivery capability. The results presented here support the conclusion that small CPPs (<30 amino acids) are sufficient to translocate

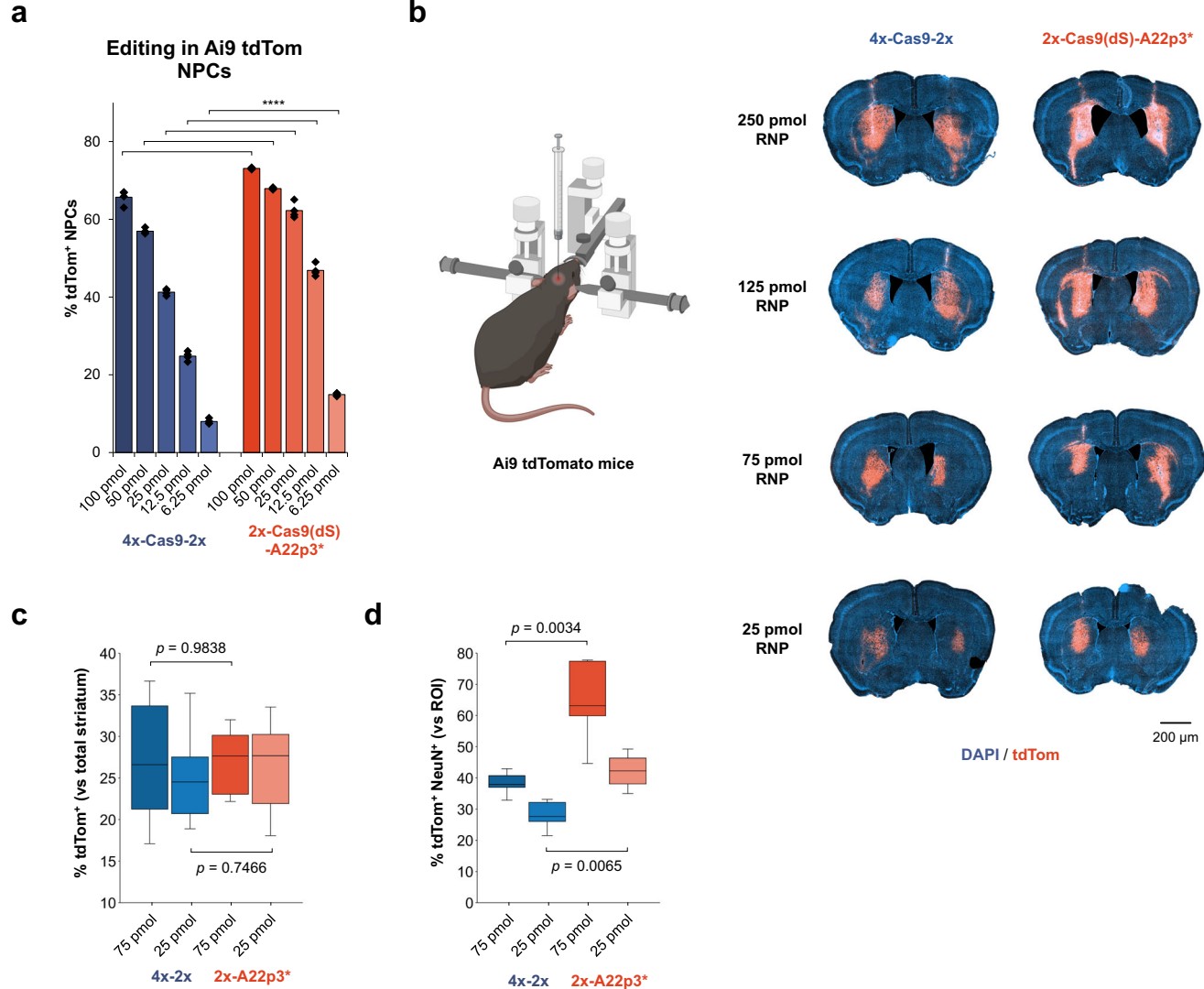

**Fig. 4 | Comparison of SV40 NLS-assisted and A22p peptide-assisted delivery of SpyCas9 for gene editing in neural cells in vitro and in vivo. a** Quantification of tdTom⁺ NPCs based on the direct delivery of RNPs to NPCs in vitro. $n = 4$ for each group, data are presented as mean values with individual data points. Statistical analyses (unpaired $t$ test) were performed, and $p$-values (****$p < 0.0001$) were indicated with each set of quantification. **b** Comparison of the gene editing activities of cell-permeable Cas9 RNPs in Ai9 mouse brain based on the injection of different RNP dosages. **c** Editing volumes in the striatal tissue. $n = 6$ for each group. **d** Co-expression of tdTomato and NeuN quantified per regions of interest (ROIs), e.g., edited area per hemisphere. $n = 4$ for the group of 2x-A22p3*, $n = 5$ for the other groups. Statistical analyses (unpaired $t$ test) were performed for (**c**, **d**) and $p$ values were indicated with each set of quantification. Data are presented in box plots for (**c**, **d**) where the lower bound of the lower whisker shows the minimum, the lower bound of the box shows the lower quartile, the center of the box shows the median, the upper bound of the box shows the upper quartile and the upper bound of the upper whisker shows the maximum. Images in (**b**) were created with biorender.com.

much larger Cas9 or Cas12 RNP cargo (150–190 kDa) into the cellular environment. CPP screening and fusion protein engineering identified SpyCas9 constructs containing three copies of C-terminally fused A22p peptide that permitted effective RNP delivery to neuronal cells both in culture and in vivo. Different from SV40 NLS as a purely cationic peptide previously used to deliver Cas9 RNP, peptide A22p consists of a minimally-charged but presumably pH-sensitive sequence and a highly cationic lysine-arginine tail; this design could possibly assist the cellular uptake of Cas9 RNP in different stages including initial internalization and later endosomal release. In addition, as A22p is derived from human semaphorin-3a and naturally binds to neuropilin receptors (NRPs) with high affinity, it was thus previously employed to promote the delivery of monoclonal antibodies by interacting with NPRs on tumor cells[31]. Based on our preliminary mechanistic studies (Supplementary Fig. 9), we suspect that recognition of the A22p ligand by receptor proteins on the neuron surface,

such as NRPs, may trigger the cellular uptake of Cas9 RNPs. In principle, similar strategies using other ligand peptides can be employed to engineer Cas proteins for in vivo delivery to other cell types[43].

In conclusion, cell-permeable CRISPR RNPs provide an effective strategy for delivering genome editors to neuronal cells in vitro and in vivo. The self-delivery capability of these genome editors, enabled by the fusion of cell-penetrating peptides, generated high levels of genome editing in neural progenitor cells with relatively low dosages of RNPs, meanwhile exhibiting minimal cytotoxicity compared to commonly used delivery methods, such as nucleofection (Supplementary Fig. 10). RNP-based genome editing of medically relevant genes in the brain demonstrated the potential utility of self-deliverable genome editors. This strategy could be extended to engineer RNPs for self-delivery to other cell types, providing a more generalized platform for therapeutic applications in different genetic disease models.

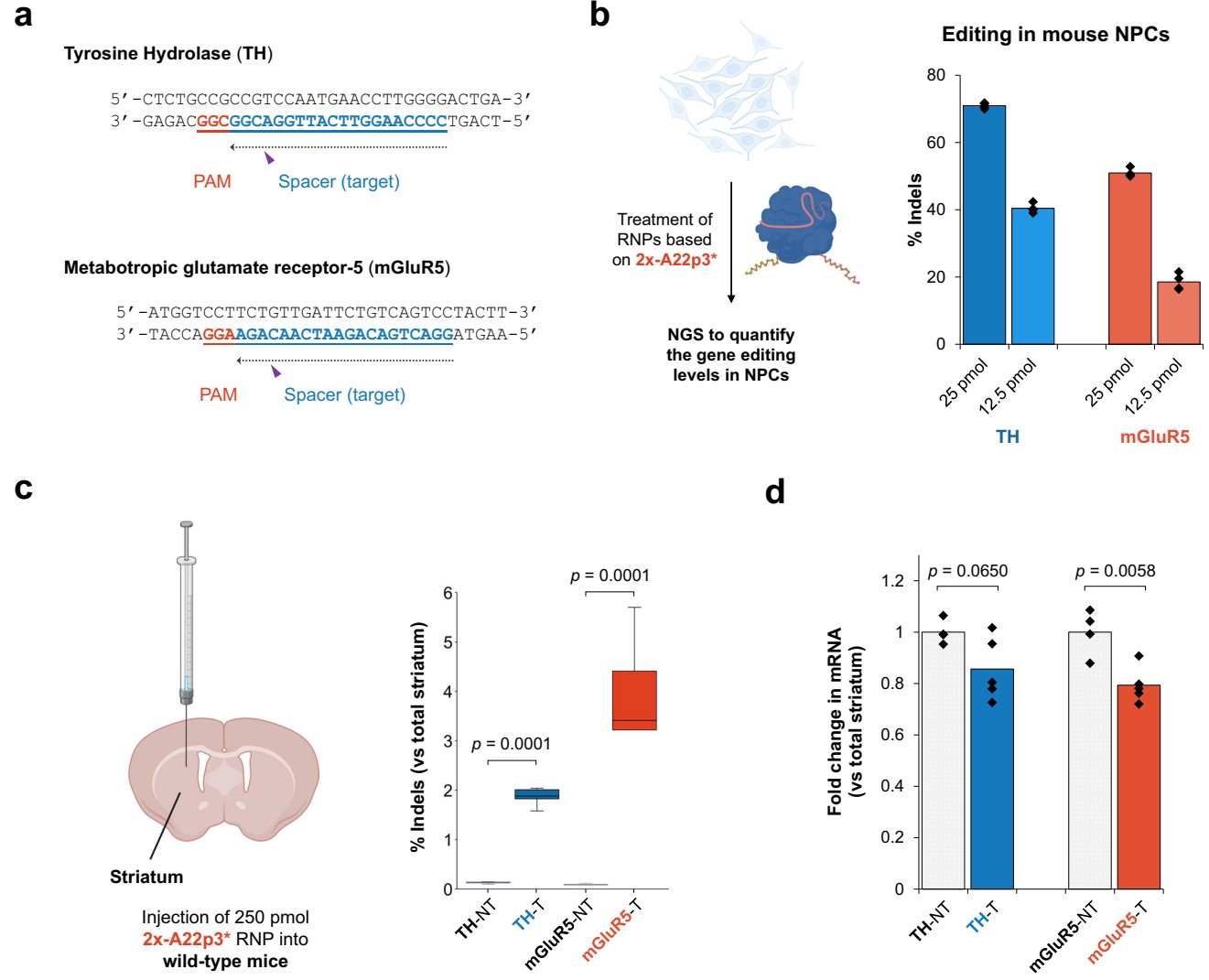

**Fig. 5 | A22p peptide-assisted delivery of SpyCas9 for gene editing at disease-relevant genomic sites. a** Schematic of the target sequences of Cas9 RNPs and the protospacer adjacent motif (PAM) for TH and mGluR5 knockout. **b** In vitro knock-out of the TH and mGluR5 genes based on direct delivery of the Cas9 RNPs. Indels quantified by NGS. $n = 4$ for each group, data are presented as mean values with individual data points. **c** In vivo knock-out of the TH and mGluR5 genes based on intraparenchymal injections of the Cas9 RNPs into mouse brains. RNPs assembled with a 1.5:1 mole ratio of sgRNA to Cas9 protein. Editing efficiency at the DNA level (indel throughout the whole striatum) quantified by NGS. $n = 4$ for the non-targeting control group and $n = 6$ for the experimental group. Data are presented in box plots where the lower bound of the lower whisker shows the minimum, the lower bound of the box shows the lower quartile, the center of the box shows the median, the upper bound of the box shows the upper quartile and the upper bound of the upper whisker shows the maximum. **d** Editing efficiency at the mRNA level (throughout the whole striatum) quantified by qPCR. $n = 4$ for the non-targeting control group and $n = 5$ for the experimental group, data are presented as median values ±s.e.m. Statistical analyses (unpaired $t$ test) were performed for (**c**, **d**), and $p$ values were indicated with each set of quantification. Images in (**b**, **c**) were created with biorender.com.

## Methods

### Ethical statement
The research presented here complies with all relevant ethical regulations. All experiments involving animals were reviewed and approved by the Animal Care and Use Committee (ACUC) at the University of California, Berkeley, prior to commencing the study.

### Plasmid construction
Plasmids used for the expression of different Cas proteins in this study were built based on a pCold vector. The inserts encoding Cas proteins contain an N-terminal CL7 tag followed by an HRV-3C protease cleavage site, and a C-terminal His$_6$ tag following another HRV-3C protease cleavage sequence. The inserts for Cas proteins with C-terminal CPPs fully exposed contain an N-terminal sequence consisting of different tags, His$_6$-CL7 followed by an HRV-3C protease cleavage site. The

cloning reactions were carried out in a 50-µl reaction containing 1 ng of template plasmid, 1.25 µl of 10 mM dNTP, and 1.25 µl of 10 µM each primer using Phusion high-fidelity DNA polymerase (New England BioLabs). After PCR, the reactions were treated with 1 µl of DpnI (New England BioLabs) for 1 h at 37 °C before gel purification. The plasmids were ligated based on Gibson assembly (New England BioLabs master mix) of plasmid backbone and insert sequences. The sequences of all the plasmid constructs were confirmed via Sanger sequencing (Barker sequencing facility, UC Berkeley) or full plasmid sequencing (Primordium).

### Nucleic acid and synthetic peptide preparation
All of the DNA oligos or dsDNA gblocks encoding CPP sequences used in this study were purchased from Integrated DNA Technologies, Inc. (IDT) and HPLC or PAGE-purified. sgRNAs were purchased from IDT or

Synthego and possess chemical modifications at 3'- or 5'-ends (See Supplementary Materials). Synthetic peptides were purchased from GenScript and HPLC-purified, giving >95% purity.

## Protein expression

All the proteins in this study were expressed in *E. coli* BL21 (DE3) cells (Sigma-Aldrich) cultured in 2x YT medium supplemented with the antibiotics of ampicillin. The cultivation was carried out at 37 °C with a shaking speed of 160 rpm after inoculation with an overnight starter culture in LB medium containing ampicillin at a ratio of 1:40. When the optical density (OD$_{600}$) of the culture reached 0.8–0.9 (generally within 3–4 h; however, the doubling time of *E. coli* that contains plasmids encoding Cas9 constructs fused with Bac7 or crotamine is roughly 1.5–2 h, because of the antimicrobial feature of the CPPs), the culture was cooled down to 4 °C on ice for 30–45 min. The expression of Cas proteins was induced by the addition of isopropyl β-D-1-thioglalacctopyranoside (IPTG) to a final concentration of 0.1 mM and incubated at 15.8–16 °C with a shaking speed of 120 rpm for 14–16 h.

To purify double-tagged proteins, the cultured cells were harvested and resuspended in lysis buffer (50 mM Tris-HCl, 20 mM imidazole, 1.2 M NaCl, 10% (v/v) glycerol, 1 mM TCEP, 0.5 mM, and cOmplete protease inhibitor cocktail tablets (Millipore Sigma, 1 tablet per 50 ml) at pH 7.5), disrupted by sonication and centrifuged at 35,000 × g for 45 min. Ni-NTA resin was treated with the supernatant at 4 °C for 60 min, washed with wash buffer-1 (lysis buffer without protease inhibitor cocktail tablet), and eluted with elution buffer (50 mM Tris-HCl, 300 mM imidazole, 1.2 M NaCl, 10% (v/v) glycerol, and 1 mM TCEP at pH 7.5) to give crude His-tagged Cas proteins. The nickel elution was then subjected to Im7-6B resin in a slow gravity column repeatedly (3–4 times). The Im7-6B resin was washed with wash buffer-2 (50 mM Tris-HCl, 1.2 M NaCl, 10% (v/v) glycerol, and 1 mM TCEP at pH 7.5) before being treated with HRV-3C protease (1% weight to crude Cas protein) for 2–2.5 h to release the Cas proteins from the CL7 and His$_6$ tags. The proteins were concentrated and analyzed by SDS-PAGE gel. Most of the proteins showed high purity (>90%) after purification with the two gravity columns. If insufficient purity was observed, heparin affinity column was used to further purify the desired proteins. The purified proteins were stored in storage buffer (25 mM NaPi, 150 mM NaCl, and 200 mM trehalose at pH 7.50) after buffer exchange. The final yields of different Cas proteins: SpyCas9-CPP, 0.5–5 mg per 1 L culture; iCas12a, 20–30 mg per 1 L culture.

The purification of endotoxin-free 2NLS-SpyCas9(dS)-A22p3* protein is slightly different. During gravity-column purification, the Ni-NTA and Im7-6B resins were washed with 10 column volumes of wash buffers containing 0.1% Triton X-114 at 4 °C to remove most of the endotoxin impurities. The tag-cleaved protein was loaded to a heparin column and washed with 80 column volumes of buffer containing 0.1% Triton X-114 at 4 °C to minimize endotoxin impurities. The protein fractions were collected, concentrated, and subjected to a further round of purification using a size-exclusion column in an endotoxin-free manner. The purified protein was stored in an endotoxin-free storage buffer (25 mM NaPi, 150 mM NaCl, and 200 mM trehalose at pH 7.50). The final yield of the desired protein: 4–5 mg per 1 L culture.

## RNP assembly and characterization

For cell culture experiments, RNPs were prepared immediately before use at a 1.2:1 mole ratio of sgRNA (Synthego or IDT) to protein (QB3 Macrolab or prepared in-house). The solution was incubated for 15–25 min at room temperature. For nucleofection, RNPs were formed at 10 µM in 10 µL of pre-supplemented buffer (Lonza P3 Primary Cell 96-well Kit, no. V4SP-3096). For direct delivery, RNPs were formed at certain concentrations (0.41 to 6.6 µM) in 15 µL of sterile storage buffer (25 mM NaPi, 150 mM NaCl, and 200 mM trehalose at pH 7.50).

For in vivo experiments targeting the tdTomato reporter gene for editing, RNPs were prepared at a 1.2:1 mole ratio of sgRNA or crRNA (Synthego or IDT) to Cas protein at 10 µM RNP concentration in an endotoxin-free phosphate buffer (25 mM NaPi, 100 mM NaCl, and 200 mM trehalose at pH 7.50) and incubated at 37 °C for 10 min. For final RNP samples with concentrations higher than 10 µM, the RNP solutions were concentrated using 50 kDa Ultra-0.5 mL Centrifugal Filter Unit (Amicon, Burlington, MA) at 14,000 × g at 4 °C until the final desired concentration (15–50 µM, minimum 80 µL volume) was reached; for final RNP samples with concentrations lower than 10 µM, the RNP solutions were diluted endotoxin-free phosphate buffer (25 mM NaPi, 100 mM NaCl, and 200 mM trehalose at pH 7.50) to the final desired concentration (5 µM, minimum 80 µL volume). RNPs were then sterile filtered by centrifuging through 0.22-mm Spin-X cellulose acetate membranes (Corning CoStar, no. 32119210) at 15,000 × g for 1 min at 4 °C. RNP solutions were collected and stored on ice for no longer than 6 h before intracranial injection.

For in vivo experiments targeting the endogenous genes for editing, RNPs were prepared at a 1.5:1 mole ratio of sgRNA (IDT) to Cas9 protein at 10 µM RNP concentration in an endotoxin-free phosphate buffer (25 mM NaPi, 100 mM NaCl, and 200 mM trehalose at pH 7.50) and incubated at 37 °C for 10 min. The RNP solutions were concentrated using 50 kDa Ultra-0.5 mL Centrifugal Filter Unit (Amicon, Burlington, MA) at 14,000 × g at 4 °C until the final desired concentration (50 µM, minimum 80 µL volume) was reached. RNPs were then sterile filtered by centrifuging through 0.22-mm Spin-X cellulose acetate membranes (Corning CoStar, no. 32119210) at 15,000 × g for 1 min at 4 °C. RNP solutions were collected and stored on ice for no longer than 6 h before intracranial injection.

The size distribution and zeta potential of RNPs (in the PBS solution) were measured using Zetasizer (version 7.13, Malvern Panalytical; He-Ne Laser, λ = 632 nm; detection angle = 173°).

## Cell line and culture conditions

NPCs were isolated from embryonic day 13.5 Ai9-tdTomato homozygous mouse brains. Cells were cultured as neurospheres at 37 °C with 5% CO$_2$ in NPC medium: DMEM/F12 (Gibco, CAT# 10565018) with GlutaMAX supplement, sodium pyruvate, 10 mM HEPES, nonessential amino acid (Gibco, CAT# 11140076), penicillin and streptomycin (Gibco, CAT# 10378016), 2-mercaptoethanol (Gibco, CAT# 21985023), B-27 without vitamin A (Gibco, CAT# 12587010), N2 supplement (Gibco, CAT# 17502048), and growth factors, bFGF (BioLegand, CAT# 579606) and EGF (Gibco, CAT# PHG0311) (both 20 ng/ml as final concentration). NPCs were passaged using MACS Neural Dissociation Kit (Papain, CAT# 130-092-628) following manufacturer's protocol. bFGF and EGF were refreshed every three days and cells were passaged every 5 days. Pre-coating with a coating solution containing poly-DL-ornithine hydrobromide (Sigma-Aldrich, CAT# P8638), laminin (Sigma-Aldrich, CAT# 11243217001), fibronectin bovine plasma (Sigma-Aldrich, CAT# F4759) was required for culturing cells in 96-well plates.

## Gene editing with NPCs

Nucleofection: 250 k NPCs cells were nucleofected with 100 pmol pre-assembled RNP (with 100 pmol ssDNA enhancer) with program code EH-100, according to the manufacturer's instructions. Lonza P3 buffer was used for the preparation of nucleofection mixtures (with a total volume of 20 µl). 10% of the nucleofected cells were transferred to 96-well plates. The culture media for NPCs was refreshed after 3 days. Cells were harvested for analysis after further incubation at 37 °C for 2 days.

Direct RNP delivery: 5k NPCs/well were seeded in 96-well plates 40–48 h prior to RNP treatment. RNPs were prepared in the storage buffer (25 mM NaPi, 150 mM NaCl, and 200 mM trehalose at pH 7.50) and added to the NPC cultures with a volume of 15 µl/well (total volume: ~120 µl/well; RNP loading/well: 6.25–100 pmol). The culture media was refreshed 2 days after RNP treatment. Cells were harvested for analysis after further incubation at 37 °C for 3 days.

## Flow cytometry

Cell fluorescence was assayed on an Attune NxT acoustic focusing cytometer (Thermo Fisher Scientific) equipped with 554 nm excitation laser and 585/16 emission filter (tdTomato). Data were analyzed using Attune Cytometric Software v5.1.1.

## Next-generation sequencing

Edited cells were harvested and treated with Quick Extraction solution (Epicentre, Madison, WI) to lyse the cells (65 °C for 20 min and then 95 °C for 20 min). Amplicons of genomic targets were PCR-amplified in the presence of corresponding primers. The PCR products were purified with magnetic beads (Berkeley Sequencing Core Facility) before being subjected to next-generation sequencing (NGS) with MiSeq (Illumina) at 2 × 300 bp with a depth of at least 20,000 reads per sample. The sequencing reads were subjected to CRISPResso2 (https://github.com/pinellolab/CRISPResso2) to quantify the levels of indels.

## Stereotaxic infusion of Cas9 RNPs

Ai9 or wild-type mice (Jackson Laboratory, Bar Harbor, ME) were group housed at the University of California, Berkeley, with a 12-h light-dark cycle and allowed to feed and drink ad libitum. Housing, maintenance, and experimentation of mice used in the study were carried out with strict adherence to ethical regulations set forth by the Animal Care and Use Committee (ACUC) at the University of California, Berkeley. Cas9-RNPs were prepared on-site for injection into male and female mice aged between 2 and 5 months. All tools were autoclaved and injected materials were sterile. Mice were anesthetized with 2% isoflurane, given pre-emptive analgesics, and arranged on an Angle Two Stereotactic Frame (Leica, Nussloch, Germany). The incision area was swabbed with three alternating wipes of 70% ethanol and betadine scrub with sterile applicators before performing minimally damaging craniotomies. The stereotaxic surgery coordinates used for targeting the striatum, relative to bregma, were +0.74 mm anteroposterior, ±1.90 mm mediolateral, and –3.37 mm dorsoventral. Bilateral convection-enhanced delivery (CED) infusion of Cas9 RNPs (5–50 µM) was performed with a syringe pump to deliver 5 µL at 0.5 µL per minute (Model 310 Plus, KD Scientific, Holliston, MA) with a step or non-step cannula. Post-infusion, the syringes were left in position for 2 min before slow removal from the injection site, which was then cleaned, sutured, and surgically glued. Throughout the procedure, mice were kept at 37 °C for warmth and Puralube Vet Ointment (Dechra, Northwich, England, NDC# 17033-211-38) was applied to the outside of the eyes. Mice were allowed to fully recover before being transferred back to their housing. Recovery weight following all procedures was monitored daily for 1 week and mice were housed without further disruption for various time periods until tissue collection.

## Tissue collection and immunostaining

At the defined study endpoints (14 days post-injection), mouse tissues were perfused with 10 mL of cold PBS and 5 mL of 4% paraformaldehyde (PFA) (Electron Microscopy Sciences, CAT# 15710). Brains were post-fixed overnight in 4% PFA at 4 °C, rinsed 3× with PBS, and then cryoprotected in 10% sucrose in PBS solution for approximately 3 days. Brains were embedded in an optimal cutting temperature medium (ThermoFisher, CAT# 23-730-571) and stored at –80 °C. Brains were cut at 20–35-mm-thick sections using a cryostat (Leica CM3050S) and transferred to positively charged microscope slides. For immunohistochemical analysis, tissues were blocked with solution (0.3% Triton X-100, 1% bovine serum albumin (Sigma-Aldrich, CAT# A9418), 5% normal goat serum (Sigma-Aldrich, CAT# G9023) before 4 °C incubation overnight with primary antibodies in blocking solution. The next day, tissues were washed three times with PBS and incubated with secondary antibodies for 1 h at room temperature. After three PBS washes, samples were incubated with DAPI solution (0.5 mg/mL, Roche Life-Science, Penzberg, Germany) as a DNA fluorescence probe for 10 min,

washed three times with PBS, submerged once in deionized water, and mounted with glass coverslips in Fluoromount-G slide mounting medium (SouthernBiotech, Birmingham, AL). Primary antibodies included mouse monoclonal anti-NeuN (1:500, Millipore Sigma, CAT# ABN78) and secondary antibodies included goat anti-rabbit 488 (1:500, ThermoFisher, CAT# A32731).

## Fluorescent imaging and image quantification

Whole-brain sections were imaged and stitched using the automated AxioScanZ1 (Zeiss, Oberkochen, Germany) with a 20× objective in the DAPI and tdTomato channels. Images generated from slide scanning were viewed in ZenLite software (v.3.6 blue edition) as CZI files. Images were then exported to Qu-Path (v.0.3.2) for quantification by authors blinded to the sample identity. Immunostained cells and tissues were imaged on the Stellaris 5 confocal microscope (Leica) with a 10× or 25× water immersion objective to capture data in DAPI, tdTomato, and FITC channels. Approximately four to six z-stack images were captured and stitched per hemisphere for quantification of NeuN with tdTomato at 1024 × 1024 pixel resolution with a scanning speed of 100–200.

Measurements of striatal editing by volume were conducted using QuPath software (v.0.3.2) from images obtained from the Zeiss AxioscanZ1. In brief, ROIs were drawn to outline the border of each striatum and the inner area of tdTomato editing using the polygon tool to create annotations. All coronal plane areas were automatically calculated. Dorsoventral coordinates (relative to bregma) were then estimated in millimeters by consulting the Mouse Brain Atlas (C57BL/6 J Coronal). Approximate tissue volume was calculated by averaging outlined areas between consecutive sections to represent the mean area across a dorsoventral segment and multiplying by the difference in dorsoventral coordinates. Edited striatal volumes were then divided by total striatal volumes to obtain percent editing.

Cell-type-specific measurements were conducted using QuPath software (v.0.3.2) on images obtained from Stellaris 5 z stack maximal projections. ROIs were again drawn around areas of observed tdTomato editing, using the polygon tool to create a single annotation per image. Cell count calculations were performed using the "Cell Detection" and "Positive Cell Detection" tools, adjusting "Cell Mean" thresholds accordingly for each channel and image.

## DNA/protein extraction from brain tissue slices

Brains were collected at 14 days for DNA and protein analysis. In brief, mouse tissues were perfused with cold PBS. Brains were harvested and cut into 2-mm sections using a matrix around the injection site (Zivic Instruments, Pittsburgh, PA). The slices were transferred onto chilled glass slides and further trimmed to approximately 30 mg tissue weight (1–1.25 mm wide × 2 mm long). Tissues were flash-frozen in liquid nitrogen and then stored at –80 °C until processing. DNA, RNA, and protein were collected from tissues using the AllPrep DNA/RNA/protein Mini Kit (QIAGEN, Venlo, the Netherlands, CAT# 80004) according to the manufacturer's instructions. In brief, brains were homogenized in 1.5-mL tubes with a disposable pestle directly in RLT lysis buffer supplemented with 2-mercaptoethanol, then passed through Qiashredder columns before adding directly to the DNA and RNA binding columns. DNA was eluted in 100 mL of EB and RNA was eluted in 40 mL RNAse-free water. Concentrations of nucleic acids were measured using a NanoDrop spectrophotometer and samples were stored at –20 °C. The RNA-binding flow-through was treated with buffer APP to precipitate protein, which was dissolved in 5% SDS buffer and further denatured at 95 °C.

## cDNA synthesis and qPCR assay

Complementary DNA (cDNA) was synthesized using the First Strand cDNA Synthesis Kit (New England BioLabs, Ipswich, MA) according to the manufacturer's instructions. Briefly, 500 ng of total RNA was mixed with 2 µL of oligo-dT primer (50 µM) in 8 µL of reaction mixture,

incubated at 65 °C for 5 min and quenched promptly on ice. Then, 10 μL of 2x ProtoScript II Reaction Mix and 2 μL of 10x ProtoScript II Enzyme Mix were added into the reaction mixture. The reaction mixture was incubated at 42 °C for 1 h and used for quantitative PCR (qPCR). qPCR was performed with 1 μL of synthesized cDNA in the 2x Maxima SYBR green /ROX qPCR Master Mix (Thermo Scientific, Emeryville, CA) using C1000 Touch Thermal Cycler (BIO-RAD, Hercules, CA). Quantification of the genes of interest was calculated as fold change to GAPDH expression. All qPCR were performed in triplicates. Primers for target genes were designed using the Primer3 program (https://primer3.ut.ee/) spanning an intron between two exons (with regard to genomic sequences) with an expected qPCR product of approximately 200 bp or less.

## Statistics and reproducibility

The data presented in bar graphs and box and whisker plots are averages across multiple biological replicates with individual data points included or error bars represented as the standard deviation. Sample sizes are indicated in the text and figure legends. When comparing two groups with normal distribution, an unpaired Student's $t$ test was performed in Prism 9 (GraphPad Software v.9.4.1). $p \leq 0.05$ was considered significant. No statistical method was used to predetermine the sample size. No data were excluded from the analyses. The experiments were not randomized. The investigators were not blinded to allocation during experiments and outcome assessment.

## Reporting summary

Further information on research design is available in the Nature Portfolio Reporting Summary linked to this article.

## Data availability

Protein, peptide, and RNA sequences in this study are available in the supplementary materials. Plasmid sequences, sequencing data, and raw images are available through Dryad (https://doi.org/10.5061/dryad.mkkwh716m). Next-generation sequencing data are available on NCBI under accession code "PRJNA1073688". Source data are provided with this paper.

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

## Acknowledgements

We thank members of the Doudna lab and the Innovative Genomics Institute for helpful discussions. We would also like to acknowledge Ms. Netravathi Krishnappa (NGS Core Operations Manager and Sequencing Specialist, Center for Translational Genomics, Innovative Genomics Institute, UC Berkeley) for NGS. K.C. is an Additional Ventures Awardee of the Life Sciences Research Foundation; J.A.D. is an Investigator of the Howard Hughes Medical Institute (HHMI). This project was supported by HHMI and Apple Tree Partners, NSF (2334028) and NIH (U19NS132303). In addition to these sources, research in the Doudna lab is supported by NIH/NIAID, NIH/NINDS, DOE, Hampton University Summer Undergraduate Research Program, Mr. Li Ka Shing, Emerson Collective and the Innovative Genomics Institute (IGI).

## Author contributions

Conceptualization: K.C. and J.A.D.; experimental studies: K.C., E.C.S., M.H.K., B.X., and R.A.; data analysis: K.C., E.C.S., M.H.K., R.A., and M.T.; supervision: J.A.D.; manuscript writing: K.C. and J.A.D. with input from all authors.

## Competing interests

The Regents of the University of California have filed patent applications for self-deliverable CRISPR RNPs under WO 2024/011176, where J.A.D. and K.C. are inventors. J.A.D. is a cofounder of Caribou Biosciences, Editas Medicine, Scribe Therapeutics, Intellia Therapeutics, and Mammoth Biosciences. J.A.D. is a scientific advisory board member or consultant for Vertex, Caribou Biosciences, Intellia Therapeutics, Scribe Therapeutics, Mammoth Biosciences, Algen Biotechnologies, Felix Biosciences, The Column Group, and Inari. J.A.D. is Chief Science Advisor to Sixth Street, a Director at Altos, Johnson & Johnson and Tempus, and she has research projects sponsored by Apple Tree Partners and Roche. The remaining authors declare no competing interests.
