## [Peer Review File · Nature Communications]

REVIEWER COMMENTS

Reviewer #1 (Remarks to the Author):

The delivery of CRISPR ribonucleoproteins (RNPs) by fused cell-penetrating peptides (CPP) for genome editing in vitro and in vivo was reported, yielding reduced off-target and immunogenic effects. These engineered self-deliverable RNPs were achieved by screening of CPPs fused to CRISPR-Cas9 protein, identifying a C-terminal Cas9 fusion with three copies of A22p, derived from human semaphorin-3a, as most efficient for genome editing in vitro and in vivo, enabling to edit 4 clinically relevant genes in the mouse brain. This strategy has high impact⁵ since it could be extended to engineer RNPs for self-delivery to other cell types for therapeutic applications in different genetic disease models.

The work supports the conclusions and claims.

The methodology is sound, the work meets the expected standards in the field.

Enough detail is provided in the methods for the work to be reproduced.

Reviewer #2 (Remarks to the Author):

This manuscript by Chen et al. focuses on engineering of RNPs for cell penetration through fusion with CPPs. By screening a collection of CPPs, the authors identified a fusion that could be dispersed over a large volume in the brain and edit neuronal cells with high efficiency. While the development of self-deliverable RNPs is of great significance, the overall novelty of this work appears limited, as similar work has been reported by other research groups (Zhang et al. *Nature Biotechnology*, 2023). Additionally, there are a few other points that need to be addressed.

1. The study primarily concentrated on screening different CPP-fusion formulations for cell penetration without conducting a thorough characterization of the other properties of these formulations. Since CPPs known to be toxic to both host *E. coli* and mammalian cells, it is crucial to characterize the CPP-Cas fusions for parameters such as yield, charge, and cytotoxicity.
2. Some key data, such as these shown in Figs. 2d, Fig. 3, and 5b, were presented without statistical analysis.
3. The authors proposed that “recognition of the A22p ligand by receptor proteins on neurons may trigger the cellular uptake of Cas9 RNPs”. Since this work focuses on the development of RNPs to neurons, the authors might want to provide experimental evidence to support the speculation.

4. Sequences of some peptides were repeatedly shown in two tables (Fig. 2b, Fig. S1) in the same manuscript.
5. Page 25, “performed with a syringe pump to deliver 5 mL at 0.5 mL per minute”. It might be 5 uL not 5 mL.

Reviewer #3 (Remarks to the Author):

In this manuscript, the authors developed self-deliverable Cas9 and Cas12 ribonucleoproteins (RNPs) for genome editing in the brain. The authors constructed a series of RNPs to screen cell-penetrating peptides fused CRISPR nucleases and identified optimal constructs. The study design is very logical and the data overall support their conclusions well. However, several issues need to be addressed to be published in Nature Communications.

1. In the earlier study (Nat Biotechnol 35, 431–434 (2017)), the construct 4NLS-Cas9-2NLS was found optimal for genome editing in the brain. Why did the authors decide to fuse 2 NLS on the N-terminus of Cas9 instead of 4 NLS?
2. The fusion of amphiphilic and/or cationic peptides may induce self-assembly of Cas9 RNP. The sizes and zeta-potentials of such RNPs should be characterized by dynamic light scattering.
3. For brain genome editing, it is also important to study other cell types including microglia and astrocytes. The in vivo editing events in these cell types need to be investigated.
4. It is found that several graphs do not have statistical analyses (e.g., Figures 1C, 2D, 3B-F, 4A, 5B, 5D, etc.).
5. It is found that all microscopic images in this manuscript do not have scale bars.
6. Peptide A22p showed the highest efficacy, which was attributed to enhanced cellular uptake by targeting neuropilin receptors. However, there was no discussion/study about their endosomal escape capabilities, which is needed for editing in the nucleus.

NCOMMS-23-51836-T

Response to reviewer comments

Reviewer #1 (Remarks to the Author):

The delivery of CRISPR ribonucleoproteins (RNPs) by fused cell-penetrating peptides (CPP) for genome editing in vitro and in vivo was reported, yielding reduced off-target and immunogenic effects. These engineered self-deliverable RNPs were achieved by screening of CPPs fused to CRISPR-Cas9 protein, identifying a C-terminal Cas9 fusion with three copies of A22p, derived from human semaphorin-3a, as most efficient for genome editing in vitro and in vivo, enabling to edit clinically relevant genes in the mouse brain. This strategy has high impact since it could be extended to engineer RNPs for self-delivery to other cell types for therapeutic applications in different genetic disease models.

The work supports the conclusions and claims.

The methodology is sound, the work meets the expected standards in the field.

Enough detail is provided in the methods for the work to be reproduced.

Response: We thank Reviewer 1 for the supportive comments.

Reviewer #2 (Remarks to the Author):

This manuscript by Chen et al. focuses on engineering of RNPs for cell penetration through fusion with CPPs. By screening a collection of CPPs, the authors identified a fusion that could be dispersed over a large volume in the brain and edit neuronal cells with high efficiency. While the development of self-deliverable RNPs is of great significance, the overall novelty of this work appears limited, as similar work has been reported by other research groups (Zhang et al. Nature Biotechnology, 2023). Additionally, there are a few other points that need to be addressed.

Response: We thank Reviewer 2 for the constructive comments. We have modified our introduction to cover these existing examples of Cas9 RNP delivery using peptides (refs. 22 & 23) and explain the novelty of our strategy.

1. The study primarily concentrated on screening different CPP-fusion formulations for cell penetration without conducting a thorough characterization of the other properties of these formulations. Since CPPs known to be toxic to both host *E. coli* and mammalian cells, it is crucial to characterize the CPP-Cas fusions for parameters such as yield, charge, and cytotoxicity.

Response: For the majority of Cas9-CPP constructs explored in this work, we did not observe a significant toxic effect on *E. coli* bacteria. Two CPPs, Bac7 and crotamine, extended *E. coli* doubling time (1.5-2 hours) because of their antimicrobial properties but did not affect protein expression (2-5 mg/L), except for the Cas9 construct with more than two copies of Bac7, as indicated in the main text. The information related to construct expression has been included in the part of "Materials and methods".

The toxicity of CPPs to NPCs was analyzed using different RNP doses, as shown in Fig. S10. The Cas9 constructs (4NLS-Cas9-2NLS or 2NLS-Cas9(dS)-A22p3*) had a minimal effect on NPC growth with RNP concentrations lower than 0.2 μM in the culture media but started to show low toxicity with RNP concentrations higher than 0.4 μM . In comparison, nucleofection generally led to <40% live cell recovery in the presence or absence of RNPs. Other features of A22p3*-fused Cas9 RNP were characterized, as included in Fig. S7.

2. Some key data, such as these shown in Figs. 2d, Fig. 3, and 5b, were presented without statistical analysis.

Response: We have now included all the necessary statistical analyses in the figures.

3. The authors proposed that “recognition of the A22p ligand by receptor proteins on neurons may trigger the cellular uptake of Cas9 RNPs”. Since this work focuses on the development of RNPs to neurons, the authors might want to provide experimental evidence to support the speculation.

Response: We speculated that the NRP-1 receptor may be involved in cellular uptake of Cas9 RNPs based on previous studies (ref. 31). To test this, we evaluated RNP delivery using NPCs with NRP-1 knocked down by siRNA and observed a decrease in RNP uptake and genome editing (Fig. S9). Additional uncharacterized receptors may also be involved in RNP uptake; however, more systematic analysis is needed to identify the exact receptors and is beyond the scope of the current study.

4. Sequences of some peptides were repeatedly shown in two tables (Fig. 2b, Fig. S1) in the same manuscript.

Response: We would like to keep Table S1 as it is, to provide a comprehensive summary of all CPPs used in the study.

5. Page 25, “performed with a syringe pump to deliver 5 mL at 0.5 mL per minute”. It might be 5 uL not 5 mL.

Response: We have fixed the typo in the revised manuscript.

Reviewer #3 (Remarks to the Author):

In this manuscript, the authors developed self-deliverable Cas9 and Cas12 ribonucleoproteins (RNPs) for genome editing in the brain. The authors constructed a series of RNPs to screen cell-penetrating peptides fused CRISPR nucleases and identified optimal constructs. The study design is very logical and the data overall support their conclusions well. However, several issues need to be addressed to be published in Nature Communications.

Response: We thank Reviewer 3 for the supportive comments.

1. In the earlier study (Nat Biotechnol 35, 431–434 (2017)), the construct 4NLS-Cas9-2NLS was found optimal for genome editing in the brain. Why did the authors decide to fuse 2 NLS on the N-terminus of Cas9 instead of 4 NLS?

Response: In the cited study, four NLS copies at the N-terminus of Cas9 were designed to function as the cell-penetrating motif, and two NLS copies at the C-terminus of Cas9 were designed for nuclear localization, although the actual functions of these NLS peptides may be different. In the current study, the two N-terminal NLS copies were designed for nuclear localization, which would allow CPP addition at the C-terminus of Cas9. Based on our experimentation, two NLS copies at the N-terminus of Cas9 are sufficient to localize RNPs in the nucleus.

2. The fusion of amphiphilic and/or cationic peptides may induce self-assembly of Cas9 RNP. The sizes and zeta-potentials of such RNPs should be characterized by dynamic light scattering.

Response: We assessed the size distribution and zeta potential of A22p3⁺-fused Cas9 RNP in comparison to peptide-free Cas9 RNP; results are now included in Fig. S7. We noticed a slight increase in the zeta potential from peptide-free RNP (-4.01±0.44 mV) to A22p3⁺-fused RNP (-2.61±0.46 mV), presumably because of the additional positive charges from NLS and A22p peptides which make the entire RNP less negatively charged. We also observed an increase in the average particle size from peptide-free RNP (d=11.4 nm) to A22p3⁺-fused RNP (d=15.8 nm), which is more substantial than the change in molecular weight (191 kDa vs 202 kDa). The size distribution remained identical for each RNP after the RNPs were incubated at room temperature (22 °C) overnight for 14 hours. We thus suspect the A22p3⁺-fused RNP may self-assemble into some reversible oligomeric states, which may more effectively trigger cellular uptake.

3. For brain genome editing, it is also important to study other cell types including microglia and astrocytes. The in vivo editing events in these cell types need to be investigated.

Response: Based on our preliminary analysis by immunofluorescence, brain editing events occurred primarily in neurons. Other cell types including microglia and astrocytes were much less frequently edited, consistent with previous studies using cell-permeable, NLS-rich Cas9 RNPs for brain editing (ref. 20 and 21).

4. It is found that several graphs do not have statistical analyses (e.g., Figures 1C, 2D, 3B-F, 4A, 5B, 5D, etc.).

Response: We have now included statistical analyses in these figures.

5. It is found that all microscopic images in this manuscript do not have scale bars.

Response: We have now included scale bars in the microscopic images.

6. Peptide A22p showed the highest efficacy, which was attributed to enhanced cellular uptake by targeting neuropilin receptors. However, there was no discussion/study about their endosomal escape capabilities, which is needed for editing in the nucleus.

Response: We have revised the text to include discussion of how peptide A22p may promote the endosomal release of RNPs: “Different from SV40 NLS as a purely cationic peptide previously used to deliver Cas9 RNP, peptide A22p consists of a minimally-charged but presumably pH-sensitive sequence and a highly cationic lysine-arginine tail; this design could possibly assist the cellular uptake of Cas9 RNP in different stages including initial internalization and later endosomal release.”

REVIEWERS' COMMENTS

Reviewer #1 (Remarks to the Author):

The concerns have been addressed in the revision

Reviewer #2 (Remarks to the Author):

The authors well addressed the reviewer's concerns.

Reviewer #3 (Remarks to the Author):

The authors have adequately addressed all of my comments.